

# *Ab initio* electron-lattice downfolding: Potential energy landscapes, anharmonicity, and molecular dynamics in charge density wave materials

Arne Schobert[1,2,3], Jan Berges[2,4], Erik G. C. P. van Loon[5], Michael A. Sentef[1,2,6,7], Sergey Brener[3], Mariana Rossi[6*] and Tim O. Wehling[3,8†]

**1** Institut für Theoretische Physik, Universität Bremen, D-28359 Bremen, Germany
**2** Bremen Center for Computational Materials Science and MAPEX Center for Materials and Processes, Universität Bremen, D-28359 Bremen, Germany
**3** I. Institute of Theoretical Physics, Universität Hamburg, D-22607 Hamburg, Germany
**4** U. Bremen Excellence Chair, Universität Bremen, D-28359 Bremen, Germany
**5** NanoLund and Division of Mathematical Physics, Department of Physics, Lund University, SE-22100 Lund, Sweden
**6** Max Planck Institute for the Structure and Dynamics of Matter, Center for Free Electron Laser Science (CFEL), D-22761 Hamburg, Germany
**7** H. H. Wills Physics Laboratory, University of Bristol, Bristol BS8 1TL, United Kingdom
**8** The Hamburg Centre for Ultrafast Imaging, D-22761 Hamburg, Germany

⋆ mariana.rossi@mpsd.mpg.de , † tim.wehling@uni-hamburg.de

## Abstract

The interplay of electronic and nuclear degrees of freedom presents an outstanding problem in condensed matter physics and chemistry. Computational challenges arise especially for large systems, long time scales, in nonequilibrium, or in systems with strong correlations. In this work, we show how downfolding approaches facilitate complexity reduction on the electronic side and thereby boost the simulation of electronic properties and nuclear motion—in particular molecular dynamics (MD) simulations. Three different downfolding strategies based on constraining, unscreening, and combinations thereof are benchmarked against full density functional calculations for selected charge density wave (CDW) systems, namely $1H\text{-}TaS_2$, $1T\text{-}TiSe_2$, $1H\text{-}NbS_2$, and a one-dimensional carbon chain. We find that the downfolded models can reproduce potential energy surfaces on supercells accurately and facilitate computational speedup in MD simulations by about five orders of magnitude in comparison to purely *ab initio* calculations. For monolayer $1H\text{-}TaS_2$ we report classical and path integral replica exchange MD simulations, revealing the impact of thermal and quantum fluctuations on the CDW transition.

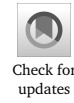



## 1 Introduction

The coupling of electronic and nuclear degrees of freedom is an extremely complex problem of relevance to multiple branches of the natural sciences, ranging from quantum materials in and out of thermal equilibrium [1–6] to chemical reaction dynamics [7,8]. Long-standing problems include the simulation of coupled electronic and nuclear degrees of freedom for large systems and large time scales, in excited states of matter or systems with strong electronic correlations. A central contributor to these challenges is the complexity of first-principles treatments of the electronic subsystem usually required to address real materials.

Charge density wave (CDW) materials exemplify these challenges. The bidirectional coupling between electrons and nuclei results in a phase transition, where the atoms of the CDW material acquire a periodic displacement from a high-temperature symmetric structure [1,3,9]. Understanding the characteristics of the CDW phase transitions, the emergence of collective CDW excitations, the control of CDW states, and excitation induced dynamics of CDW systems [10–19] requires typically simulations on supercells involving several hundred or thousand atoms, where eV-scale electronic processes intertwine with collective mode dynamics at the meV scale. CDW systems thus define a formidable spatio-temporal multiscale problem. Solutions to this problem can be attempted with variational techniques [20–24], which neglect certain anharmonic effects like the anharmonic phonon decay, or by trying to circumvent the multi-scale problem by scale-separation [25, 26].

Corresponding complexity reduction strategies have been developed in distinct fields: Multi-scale coarse-grained models, machine-learning models [27–31], or (density functional) tight binding potentials [32–53] have been put forward. In these methods, models are defined by fitting semiempirical or "machine learned" (neural networks, Gaussian processes, others) parameter functions to reference data often taken from density-functional theory (DFT) [54] calculations.

In the field of strongly correlated electrons, one also deals with minimal models, which typically focus on low-energy degrees of freedom: The electronic system is divided into high- and low-energy sectors. Then, the high-energy states are integrated out via field theoretical or perturbative means, leaving an effective low-energy model [55]. Methods for the derivation of model parameters include the constrained random phase approximation (cRPA) [56–62], constrained density functional perturbation theory (cDFPT) [63–67], and the constrained functional renormalization group [68–70]. The field theoretical integrating out of certain electronic states is often called "downfolding".

In this work, we demonstrate how downfolding approaches for complexity reduction on the electronic side boost the simulation of coupled electronic and nuclear degrees of freedom— in particular molecular dynamics (MD) simulations. The idea is to map the first-principles solid-state Hamiltonian onto minimal quantum lattice models, where "minimal" refers to the dimension of the single-particle Hilbert space. Three different downfolding strategies based on constraining, unscreening, and combinations thereof, are compared and demonstrated along example cases from the domain of CDW materials.

We start by introducing the first-principles electron-nuclear Hamiltonian and the minimal quantum lattice models together with the three downfolding schemes in Section 2. Potential energy surfaces resulting from the downfolded models are benchmarked against DFT for exemplary CDW systems in Section 3. MD simulations based on a downfolded model are presented in Section 4, where the CDW transition of 1H-$TaS_2$ is studied as a function of temperature, and the computational performance gain from downfolding is analyzed.

## 2 From first-principles to minimal lattice models

The general Hamiltonian of interacting electrons and nuclei in the position representation and atomic units, where in particular $m_e = e = 1$, reads

$$H_{FP} = -\sum_i \frac{\Delta_i}{2} - \sum_k \frac{\Delta_k}{2M_k} + \sum_{i<j} \frac{1}{|r_i - r_j|} + \sum_{k<l} \frac{Z_k Z_l}{|R_k - R_l|} - \sum_{ik} \frac{Z_k}{|r_i - R_k|} , \qquad (1)$$

where $r_i$ and $R_k$ are electronic and nuclear positions, $\Delta_i$ and $\Delta_k$ are the corresponding Laplace operators, and $Z_k$ and $M_k$ are atomic numbers and nuclear masses. This Hamiltonian is also called "first-principles (FP) Hamiltonian", since only fundamental laws (i.e., the Schrödinger equation, Coulomb potential, etc.) and fundamental constants (elementary charges etc.) enter. It accounts for full atomic scale and chemical details. Numerical treatments leading directly from this Hamiltonian to physical results are called "*ab initio*", cf. Fig. 1 (left).

In principle, DFT provides us with a tool to calculate the total (free) energy and forces given fixed atomic positions $R_k$ as needed for MD simulations in the Born-Oppenheimer approximation [71]. However, DFT calculations with large supercells can become prohibitively expensive (cf. Fig. 7 for benchmark calculations later in this work). As a consequence, DFT simulations of phase transitions governed by inhomogeneity effects are often very challenging. It is, thus, desirable to obtain energies and forces in a cheaper way, while remaining close to the quantum mechanical accuracy of *ab initio* simulations.

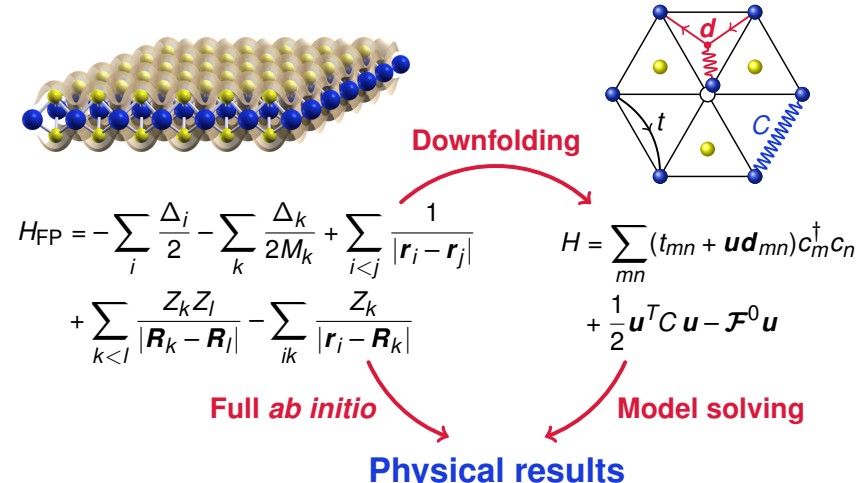

**First-principles Hamiltonian**          **Minimal lattice model**

**Downfolding**

$$H_{\mathrm{FP}} = -\sum_i \frac{\Delta_i}{2} - \sum_k \frac{\Delta_k}{2M_k} + \sum_{i<j} \frac{1}{|\boldsymbol{r}_i - \boldsymbol{r}_j|}$$
$$+ \sum_{k<l} \frac{Z_k Z_l}{|\boldsymbol{R}_k - \boldsymbol{R}_l|} - \sum_{ik} \frac{Z_k}{|\boldsymbol{r}_i - \boldsymbol{R}_k|}$$

$$H = \sum_{mn} (t_{mn} + \boldsymbol{u}\boldsymbol{d}_{mn}) c_m^\dagger c_n$$
$$+ \frac{1}{2} \boldsymbol{u}^T C \boldsymbol{u} - \boldsymbol{\mathcal{F}}^0 \boldsymbol{u}$$

**Full *ab initio***                    **Model solving**

**Physical results**

Figure 1: *Ab initio* versus *ab initio* based downfolding approaches to coupled electron-nuclear dynamics.

Here, our goal is to use a reduced low-energy electronic Hilbert space for this purpose, with only a few orbitals per unit cell, cf. Fig. 1 (right).

We thus aim to work with a lattice model

$$H = H_{\mathrm{el}} + H_{\mathrm{n}} + H_{\mathrm{el\text{-}n}}, \tag{2}$$

which consists of the low-energy electronic subsystem

$$H_{\mathrm{el}} = H_{\mathrm{el}}^0 + H_{\mathrm{el}}^1 + H_{\mathrm{DC}}, \tag{3}$$

with one-body

$$H_{\mathrm{el}}^0 = \sum_{\boldsymbol{k}n} \varepsilon_{\boldsymbol{k}n}^0 c_{\boldsymbol{k}n}^\dagger c_{\boldsymbol{k}n}, \tag{4}$$

Coulomb interaction

$$H_{\mathrm{el}}^1 = \frac{1}{2N} \sum U_{\boldsymbol{q}\boldsymbol{k}mnk'm'n'} c_{\boldsymbol{k}+\boldsymbol{q}m}^\dagger c_{\boldsymbol{k}'n'}^\dagger c_{\boldsymbol{k}'+\boldsymbol{q}m'} c_{\boldsymbol{k}n}, \tag{5}$$

and double counting ($H_{\mathrm{DC}}$) parts, the nuclear subsystem

$$H_{\mathrm{n}} = -\sum_k \frac{\Delta_k}{2M_k} + V^0(\boldsymbol{u}_1, \ldots, \boldsymbol{u}_{N_{\mathrm{n}}}), \tag{6}$$

and a coupling between the electronic and nuclear degrees of freedom

$$H_{\mathrm{el\text{-}n}} = \sum_{\boldsymbol{q}\boldsymbol{k}mn} V_{\boldsymbol{q}\boldsymbol{k}mn}(\boldsymbol{u}_1, \ldots, \boldsymbol{u}_{N_{\mathrm{n}}}) c_{\boldsymbol{k}+\boldsymbol{q}m}^\dagger c_{\boldsymbol{k}n}. \tag{7}$$

The electronic subspace is spanned by a set of low-energy single particle states $|\boldsymbol{k}n\rangle$, with $\boldsymbol{k}$ the crystal momentum, and $n$ summarizing further quantum numbers (band index, spin). $c_{\boldsymbol{k}n}^\dagger$ ($c_{\boldsymbol{k}n}$) are the corresponding electronic creation (annihilation) operators. $N$ is the number of $\boldsymbol{k}$ points summed over. The nuclear degrees of freedom are expressed in terms of displacements $(\boldsymbol{u}_1, \ldots, \boldsymbol{u}_{N_{\mathrm{n}}}) \equiv \boldsymbol{u} = \boldsymbol{R} - \boldsymbol{R}_0$ from a relaxed reference structure $\boldsymbol{R}_0$.

$H_{\text{el}}$ describes the low-energy electronic subsystem in the non-distorted configuration ($\boldsymbol{u}=0$) with the effective electronic dispersion $\varepsilon^0_{\boldsymbol{k}n}$ and an effective Coulomb interaction $U$. In this work, $\varepsilon^0_{\boldsymbol{k}n}$ is always taken from the DFT Kohn-Sham eigenvalues of the undistorted reference system. Whenever $U \neq 0$, a term $H_{\text{DC}}$ has to be added to avoid double counting (DC) of the Coulomb interaction already contained in the Kohn-Sham eigenvalues (see Appendix A).

$V^0$ plays the role of an effective interaction between the nuclei, or equivalently a partially screened deformation energy, which accounts for the Coulomb interaction between the nuclei and the interaction between the nuclei and the high-energy electrons not accounted for in $H_{\text{el}}$. In this work, we expand $V^0$ to second order in the atomic displacements $\boldsymbol{u}$,

$$H_{\text{def}} = V^0 = -\sum_i \mathcal{F}^0_i u_i + \frac{1}{2}\sum_{ij} u_i C_{ij} u_j \,, \tag{8}$$

where $\mathcal{F}^0$ is a force vector and $C$ a force constant matrix. The coupling between the displacements and the low-energy electronic system from Eq. (7) is expanded to first order in the displacements $\boldsymbol{u}$:

$$H_{\text{el-n}} = \boldsymbol{u}\sum_{\boldsymbol{q}\boldsymbol{k}mn} \boldsymbol{d}_{\boldsymbol{q}\boldsymbol{k}mn} c^\dagger_{\boldsymbol{k}+\boldsymbol{q}m} c_{\boldsymbol{k}n} \,. \tag{9}$$

Here, $\boldsymbol{d}_{\boldsymbol{q}\boldsymbol{k}mn} = \boldsymbol{\nabla}_{\boldsymbol{u}} V_{\boldsymbol{q}\boldsymbol{k}mn}(\boldsymbol{u})$, and $\boldsymbol{u} \cdot \boldsymbol{d}_{\boldsymbol{q}\boldsymbol{k}mn}$ plays the role of a displacement-induced potential acting on the low-energy electrons.

MD simulations are a major motivation for constructing the low-energy electronic model. These simulations are here performed at various temperatures, using an electronic model that is established based on a single DFT and density functional perturbation theory (DFPT) calculation. The effective free energy of the system at given nuclear coordinates $\boldsymbol{R} = \boldsymbol{R}_0 + \boldsymbol{u}$ is

$$F(\boldsymbol{u}) = -kT \log Z(\boldsymbol{u}) \,. \tag{10}$$

Here, the partition function $Z(\boldsymbol{u}) = \text{Tr}_{\text{el}} \exp(-\beta H)$ traces out the electronic degrees of freedom but not the nuclei. Thus, $F(\boldsymbol{u})$ plays the role of a potential energy surface, which governs the dynamics of the nuclei in Born-Oppenheimer approximation. Forces acting on the nuclei are then $\mathcal{F} = -\boldsymbol{\nabla}_{\boldsymbol{u}} F(\boldsymbol{u})$ and can be conveniently obtained using the Hellmann-Feynman theorem (see Appendix B):

$$\mathcal{F} = -\sum_{\boldsymbol{q}\boldsymbol{k}mn} \boldsymbol{d}_{\boldsymbol{q}\boldsymbol{k}mn} \langle c^\dagger_{\boldsymbol{k}+\boldsymbol{q}m} c_{\boldsymbol{k}n} \rangle \,. \tag{11}$$

$C$, $U$, and $\boldsymbol{d}$ entering the model Hamiltonian $H$ are not bare but (partially) screened quantities. The (partial) screening has to account for electronic processes not contained explicitly in $H$. Here, we consider three different schemes to determine $C$, $U$, and $\boldsymbol{d}$:

**Model I** strictly follows the idea of the constrained theories [57, 64]. In these theories, the high-energy electronic degrees of freedom are integrated out to derive the low-energy model. The parameters entering the low-energy Hamiltonian are therefore "partially screened" by the high-energy electrons. In particular, we use cRPA for the Coulomb interaction $U$ and cDFPT for the displacement-induced potential $\boldsymbol{d}$ and for the force constant matrix $C$.

**Model II** again applies $U$ from cRPA. Now, however, $\boldsymbol{d}$ and $C$ are based on the *unscreening* of the respective DFPT quantities using $U$ inspired by Ref. [72].

**Model III** considers a non-interacting low-energy system, $U = 0$. $\boldsymbol{d}$ is taken from DFPT. $C$ is obtained from unscreening DFPT. This approach is inspired by Ref. [73].

In all models, the force vector $\mathcal{F}^0$ entering $H_{\text{def}}$ in Eq. (8) is chosen to guarantee that $\mathrm{d}F/\mathrm{d}u_i|_{\boldsymbol{u}=0} = 0$, i.e., vanishing forces also in the models for the reference structure $\boldsymbol{R}_0$. The term $-\mathcal{F}^0\boldsymbol{u}$, thus, plays the role of a "force double counting correction" similar to Refs. [63,64].

Since the downfolding is done on the primitive unit cell for $\boldsymbol{u} = 0$, and we are interested in the potential energy surface for displacements on supercells, we have to map the model parameters $\varepsilon^0$, $C$, $U$, and $\boldsymbol{d}$ from the unit cell to the supercell. For displacements with the supercell periodicity, we can set $\boldsymbol{q} = 0$ in Eq. (9) and—within the random phase approximation (RPA)—also in Eq. (5) and drop the corresponding subscripts.

We have implemented this mapping for arbitrary commensurate supercells defined by their primitive lattice vectors $\boldsymbol{A}_i = \sum_j N_{ij}\boldsymbol{a}_j$ with integer $N_{ij}$ [74]. It relies on localized representations in the basis of Wannier functions and atomic displacements [75], for which the mapping is essentially a relabeling of basis and lattice vectors.

## 2.1 Unscreening in models II and III

The central idea of models II and III is to choose $C$ entering $H_{\text{def}}$ such that $\mathrm{d}^2 F/\mathrm{d}u_i\mathrm{d}u_j|_{\boldsymbol{u}=0} = C_{ij}^{\text{DFT}}$, where the latter are the DFT force constants, accessible via DFPT. In model II we additionally require that the screened deformation-induced potential and accordingly the screened electron-phonon vertex at the level of the static RPA matches the corresponding DFPT quantity.

The unscreening procedure is represented diagrammatically: The Green's function resulting from the undistorted Kohn-Sham dispersion $\varepsilon_{\boldsymbol{k}n}^0$ is shown as a black arrow line, $G \to \longrightarrow$. We use a wavy line to denote the Coulomb interaction $U \to \sim$ obtained from cRPA. The deformation-induced potential obtained from DFPT, which is by definition fully screened, is represented as a black dot, $\boldsymbol{d}^{\text{DFT}} = \boldsymbol{d}^{\text{III}} \to \bullet$.

### 2.1.1 Model II

We define the unscreened deformation-induced potential $\boldsymbol{d}^{\text{II}} \to \bullet$ (red dot) entering model II via Eq. (9) as

$$\bullet = \bullet - \sim\!\!\bigcirc\!\!\bullet, \tag{12}$$

which can be written in shorthand notation as $d^{\text{II}} = d - U\Pi d$, or explicitly as

$$\boldsymbol{d}_{\boldsymbol{k}mn}^{\text{II}} = \boldsymbol{d}_{\boldsymbol{k}mn} - \frac{1}{N}\sum_{\boldsymbol{k}'m'n'\alpha\beta}\varphi_{\boldsymbol{k}\alpha m}^*\varphi_{\boldsymbol{k}\beta n}U_{\alpha\beta}\varphi_{\boldsymbol{k}'\alpha m'}\varphi_{\boldsymbol{k}'\beta n'}^*\frac{f(\varepsilon_{\boldsymbol{k}'m'})-f(\varepsilon_{\boldsymbol{k}'n'})}{\varepsilon_{\boldsymbol{k}'m'}-\varepsilon_{\boldsymbol{k}'n'}}\boldsymbol{d}_{\boldsymbol{k}'m'n'}. \tag{13}$$

Here, $\varepsilon_{\boldsymbol{k}n}$, $\varphi_{\boldsymbol{k}\beta n}$ are the eigenvalue and -vector of band $n$ from the undistorted Wannier Hamiltonian, and $U_{\alpha\beta}$ is the cRPA Coulomb interaction in the orbital basis.

The definition in Eq. (12) implies that the static RPA screening of the deformation-induced potential in model II indeed matches the DFPT input, since

$$\bullet = \bullet + \sim\!\!\bigcirc\!\!\bullet$$

$$= \bullet + \sim\!\!\bigcirc\!\!\bullet + \sim\!\!\bigcirc\!\!\sim\!\!\bigcirc\!\!\bullet + \ldots \tag{14}$$

The force constant matrix $C = C^{\text{DFT}} - \Delta C^{\text{RPA}}$ entering model II is obtained by unscreening the DFPT fully screened force constants $C^{\text{DFT}}$ on the RPA level, i.e., we subtract the second-order response in RPA of the electronic system to the atomic displacements, as given by the bubble diagram

$$\Delta C^{\text{RPA}} = \bullet\!\!\bigcirc\!\!\bullet. \tag{15}$$

Table 1: Comparison of downfolded models.

|  | Model I | Model II | Model III |
|---|---|---|---|
| **Coulomb interaction** [Eq. (5)] | cRPA | cRPA | — |
| **Electron-phonon coupling**[1] [Eq. (9)] | cDFPT | DFPT $(\star)^*$ 🔴 | DFPT ⚫ |
| **Force constants** [Eq. (8)] | cDFPT | DFPT $(\star)^*$ | DFPT $(\star)^*$ |

[1] as in displacement-induced potential.
* $(\star)$ refers to *unscreened* quantities.

### 2.1.2  Model III

Again, we construct the total free energy to be exact in second order. As in model II, we have to subtract the unwanted second order, $C = C^{\text{DFT}} - \Delta C^{\text{III}}$. The change in the interatomic force constants for this non-interacting model is given by the bubble diagram (cf. Appendix B)

$$\Delta C^{\text{III}} = \bullet\!\!\bigcirc\!\!\bullet . \tag{16}$$

The unscreening is exact when the DFT force constants, the bubble diagram, and the free energy are evaluated at the same electronic temperature $T_{\text{DFT}}$. This electronic temperature facilitates the treatment of metals within DFT calculations. However, on the model side we have the freedom to evaluate the free energy at a different electronic temperature $T_M$. Interestingly, the resulting second order is still a very good approximation to the DFT force constants at temperature $T_M$ [73,76], as it will be demonstrated in this work.

This completes the definitions of models I, II, and III, which are also summarized in Table 1. In the following, we will explain and demonstrate the downfolding according to models I–III along the example case of monolayer 1H-TaS$_2$.

## 3  CDW potential energy landscapes in 1H-TaS$_2$: DFT vs downfolding

Monolayer 1H-TaS$_2$ exhibits a $3 \times 3$ CDW [77–79], where atoms are displaced from their symmetric positions as illustrated in Fig. 2a. Coupling between electrons within the low-energy subspace (highlighted in Fig. 2b) and the lattice distortions $\boldsymbol{u}$ is responsible for the $3\times3$ CDW instability [66]. Hence, we choose these three bands to span the low-energy subspace of electrons in the Hamiltonian $H$.

We present practical calculations using downfolded models I–III and benchmark the resulting potential energy landscapes against full DFT calculations. Details of the DFT calculations are presented in Appendix C. The energy landscapes will be illustrated along the displacement direction of the CDW distortion: $\boldsymbol{u} = \alpha(\boldsymbol{R}_{\text{CDW}} - \boldsymbol{R}_0)$. Here, $\boldsymbol{R}_0$ is the symmetric relaxed structure, and $\boldsymbol{R}_{\text{CDW}}$ is the CDW structure as obtained by DFT. $\alpha$ plays the role of a scalar coordinate, where by construction $\alpha = 0$ yields the symmetric state and $\alpha = 1$ the CDW displacement pattern. Note, however, that the models readily yield the full energy landscape for arbitrary displacements.

**Model I** starts with partially screened force constants $C$ from cDFPT in $H_{\text{def}}$, which exclude screening processes taking place within the low-energy electronic target space highlighted in Fig. 2b. The "bare" harmonic potential energy versus displacement curves resulting from $H_{\text{def}}$ (dark gray cDFPT parabola) is compared to full DFT total energy calculations (crosses) in Fig. 2c. The upward opened cDFPT parabola shows that the CDW lattice instability is induced by the electrons of the target subspace, in accordance with Ref. [66].

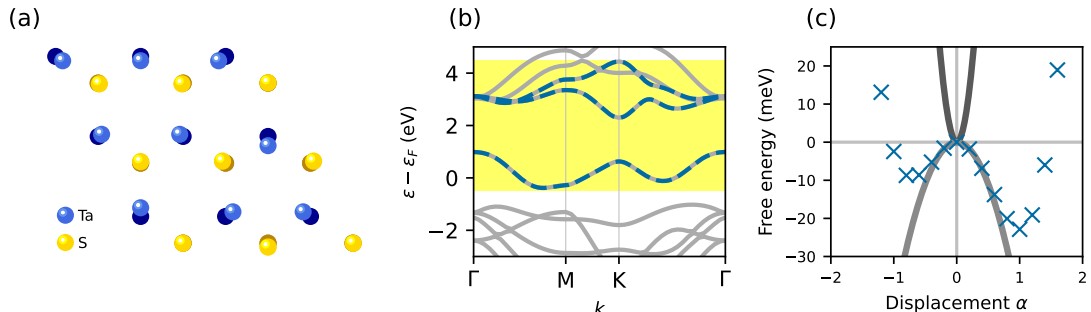

Figure 2: (a) Crystal structure of the $3 \times 3$ CDW in 1H-TaS$_2$ (displacements are increased by a factor of 5 for visibility). (b) Electronic bands of 1H-TaS$_2$ from DFT (gray) and Wannier bands (blue dashed), which span the cDFPT active subspace highlighted in yellow. (c) Born-Oppenheimer potential energy surface from DFT for the $3 \times 3$ CDW in 1H-TaS$_2$ (blue crosses). Its negative curvature matches the DFPT parabola (light gray curve). The cDFPT parabola, which is not screened by the active subspace electrons, is opened upward (dark gray curve).

We account for density-density type Coulomb matrix elements in $H_{el}$, which we obtain from cRPA, and solve the resulting model Hamiltonian $H$ for the potential energy landscape $F(\boldsymbol{u})$ in Hartree approximation. See Appendix A for a detailed description of the Hartree calculations. The resulting total (free) energy versus displacement curve is compared to DFT in Fig. 3a. Model I generates an anharmonic double-well potential and thus features a CDW instability like DFT, which is qualitatively reproduced. Nevertheless, there is some deviation of model I from DFT, which originates mainly from the harmonic term. In comparison to DFT, model I and its subsequent Hartree solution involve two additional approximations, which could be responsible for the deviations to second order: neglecting non density-density type Coulomb terms, and neglecting exchange-correlation effects.

**Model II** suppresses deviations from the DFT potential energy landscape to second order in $\boldsymbol{u}$ by construction: Since the fully screened deformation energy from DFPT agrees with the DFT energy versus displacement curve (see Fig. 2b), as it must be, also the solution of the downfolded model II matches DFT to second order in the displacement (Fig. 3b). The overall match between the downfolded model II and DFT is clearly much better than for model I and indeed almost quantitative also at displacements $|\alpha| > 1$, where anharmonic terms are substantial.

Also **model III**, which involves non-interacting electrons coupled to lattice deformations via fully screened DFPT displacement-induced potentials, recovers the DFT potential energy vs displacement curve for the $3 \times 3$ CDW distortion in 1H-TaS$_2$ almost quantitatively (Fig. 3c) and even slightly better than model II.

We also applied downfolded model III to monolayer 1T-TiSe$_2$, a one-dimensional carbon chain, and monolayer 1H-NbS$_2$ as examples of further CDW materials. The resulting potential energy landscapes in Fig. 4 show the agreement between DFT and the downfolded model. Hence, model III captures the most important anharmonicities in these cases. CDWs are especially, but not exclusively, found in low-dimensional systems. As a consequence, we focussed on low-dimensional materials for this benchmark. However, the downfolding formalism is independent of dimensionality.

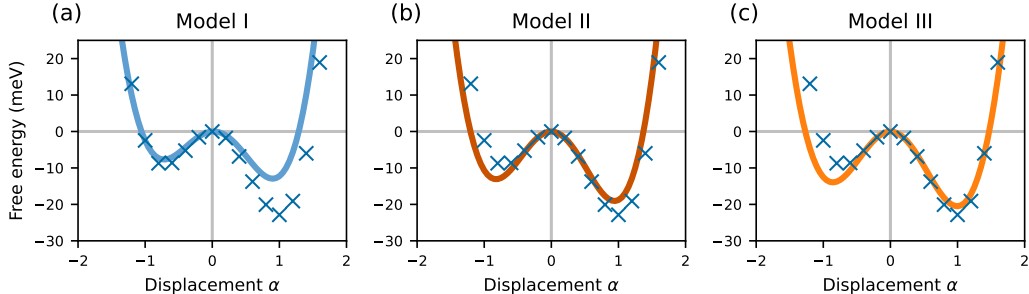

Figure 3: Free energies of the $3 \times 3$ CDW mode in 1H-TaS$_2$ from DFT (blue crosses) and downfolded models. (a) Interacting model with partially screened quantities from constrained theories cRPA and cDFPT (start from cDFPT parabola). (b) Interacting model with partially screened quantities from unscreening (start from DFPT parabola). (c) Non-interacting model with fully screened quantities (start from DFPT parabola).

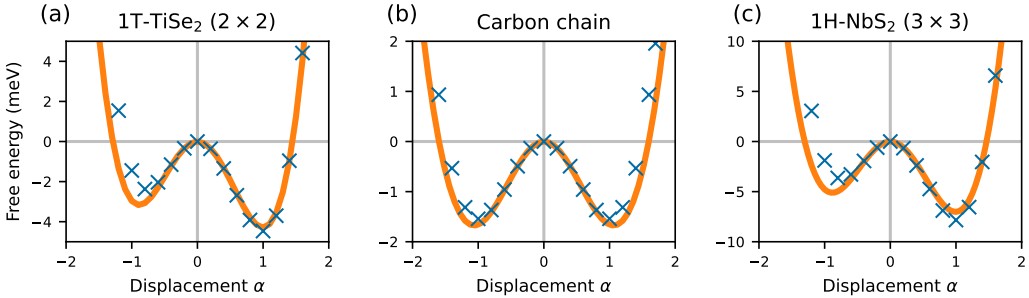

Figure 4: Free energies of (a) the $2 \times 2$ CDW in 1T-TiSe$_2$, (b) the CDW in the carbon chain, and (c) the $3 \times 3$ CDW in 1H-NbS$_2$. The blue crosses are data points from DFT and the orange curves are the model III results.

## 3.1 Influence of Wannier orbitals and electronic Hilbert space dimension

Since the electronic Hamiltonian [Eq. (3)] is represented via Wannier functions, we have a certain freedom of choice. From a computational standpoint, we are aiming for a maximal reduction of the dimension of the single-particle Hilbert space, while maintaining a reasonable level of accuracy. Thus, the natural question arises: How many and which Wannier orbitals to choose to create the single-particle Hilbert space?

For 1H-TaS$_2$, we compare a "minimal" and a "maximal" model involving, respectively, three and eleven Wannier orbitals per unit cell: In the case of three orbitals, there are three $d$-type orbitals on the Ta atom ($d_{z^2}, d_{x^2-y^2}, d_{xy}$), and in the case of eleven orbitals, there are five $d$-type orbitals on the Ta atom ($d_{z^2}, d_{xz}, d_{yz}, d_{x^2-y^2}, d_{xy}$) and three $p$-type orbitals on both S atoms ($p_x, p_y, p_z$). Note that these are the Hilbert space dimensions on the primitive unit cell. On the $3 \times 3$ supercell calculations, the dimensions are 27 and 99 respectively.

We compare the energy-displacement curves resulting from model III for both Hilbert space sizes to DFT in Fig. 5. While the results are similar in both cases, the eleven orbital model is slightly closer to full DFT than the three orbital model. In the eleven-orbital model, the displacement potentials directly induce changes in the $d$-$p$ hybridization. We speculate that anharmonicities associated with these rehybridization terms are responsible for the slightly improved accuracy of the eleven band model.

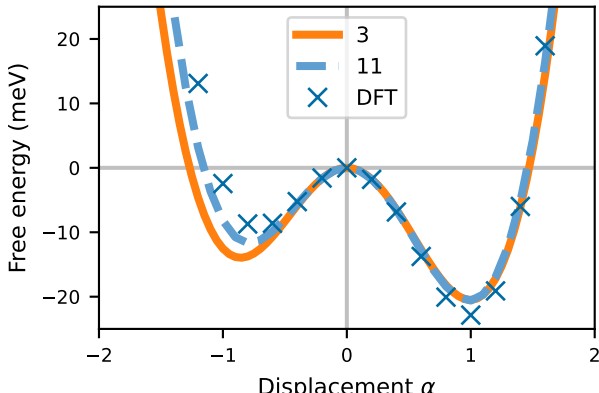

Figure 5: Free energies of the $3 \times 3$ CDW in 1H-TaS$_2$. We show DFT data points (blue crosses), model III results for three Wannier orbitals (orange solid curve), and model III results for eleven Wannier orbitals (blue dashed curve).

## 3.2 Electronically generated anharmonicities

Models I, II, and III are based on the electronic structure at the symmetric equilibrium positions of the atoms, as well as the linear response to displacements that is accessible in DFPT. By construction, models II and III guarantee agreement with the full DFT calculation at small displacements $u$, up to order $u^2$ in the energy and up to order $u$ in the electronic structure. One might wonder if these models, based on linear response, can ever be useful for the description of the distorted phase, which is necessarily stabilized by anharmonicity and terms of order $u^3$, $u^4$, and beyond.

The close match between the significantly anharmonic DFT potential energy landscapes and models II and III in Figs. 3, 4, and 5 at $|\alpha| > 1$ might thus come as a surprise. The reason behind the good match even in the anharmonically dominated region can be understood in the following sense: Linear changes in the electronic potential lead to non-linear changes in eigenvalues of the electronic Hamiltonian and therefore in the total energy. Thus, if the low-energy electrons are responsible for the anharmonicity that stabilizes the CDW, then a low-energy electronic model based on DFPT quantities has the possibility to describe this.

The emergence of electronically driven anharmonicities can be illustrated with an electronic two-level system, $H_{el}^0 = \Delta \sigma_z$, coupled linearly to a nuclear displacement $u$ through $H_{el-n} = u \cdot d \sigma_x$ [following Eq. (9)]. Here, $\sigma_i$ denote Pauli matrices, $2\Delta$ is the level-splitting and $d$ encodes the strength of the coupling of electrons to nuclear displacements as in Eq. (9). The ground state eigenvalue of $H_{el}^0 + H_{el-n}$ reads $E_0 = -\sqrt{\Delta^2 + (du)^2} \approx -\Delta \left(1 + \frac{1}{2}\left(\frac{du}{\Delta}\right)^2 - \frac{1}{8}\left(\frac{du}{\Delta}\right)^4 + \dots\right)$. Thus, electronically generated anharmonicities appear at displacements on the order $u \approx \Delta/d$. Taking the level splitting $\Delta$ as a proxy for the electronic bandwidth $W \sim \Delta$ or for the inverse of the density of states at the Fermi level $\rho \sim 1/\Delta$, we have electronically generated anharmonicities appearing at displacements on the order $u \approx W/d \approx 1/(\rho d)$. In other words, systems with strong electron-lattice coupling and high density of states at the Fermi level are expected to be domains where the linearized electron-lattice coupling preferably works. In addition, the approximation of a linearized electron-lattice coupling as in Eq. (9) has also been successfully applied to describe polaronic lattice distortions [80, 81].

This hypothesis is further corroborated by the comparison of energy-displacement curves for 1H-TaS$_2$ at different electronic smearings to those of the related system 1H-WS$_2$, in Fig. 6.

The electronic band structure of WS$_2$ [82] is very similar to the one of TaS$_2$ (see Fig. 2b) with the key difference that it has one additional valence electron per unit cell. Hence, the half-

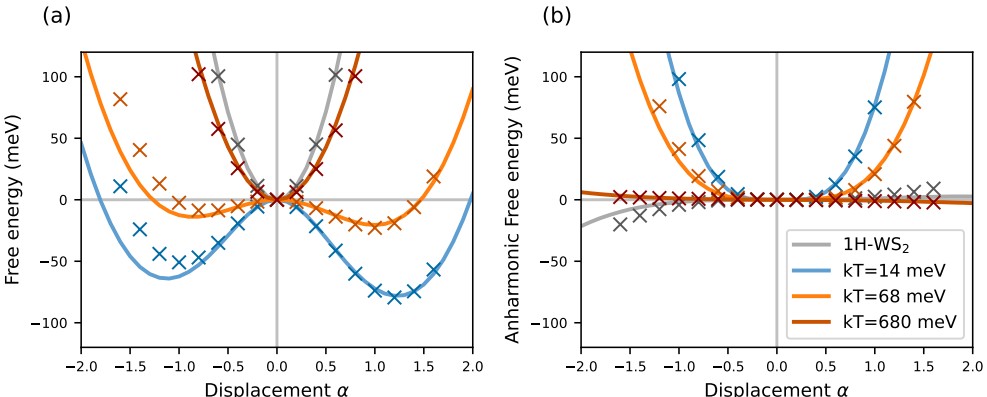

Figure 6: (a) Free energy and (b) anharmonic part of the free energy for 1H-TaS$_2$ at electronic smearings $kT = 14$ meV (blue), 68 meV (orange), 680 meV (dark red) and for 1H-WS$_2$ at smearing $kT = 68$ meV. Solid lines originate from model III and crosses from DFT. Even though the inputs for model III (see Table 1) were generated at the electronic temperature $T_{\mathrm{DFT}} = 68$ meV, we can still evaluate the free energy at higher or lower model temperatures $T_{\mathrm{M}}$ and get a good agreement with DFT.

filled conduction band of TaS$_2$ becomes completely filled in the WS$_2$ case, which renders WS$_2$ semiconducting and quenches the response of the low energy electronic system. Similarly, an increased electronic smearing/temperature quenches the response of the low-energy electronic system. Both WS$_2$ and TaS$_2$ at high smearing, are dynamically stable, which is indicated by the positive second order of the free energy in Fig. 6a. This tells us that at least the harmonic term is significantly affected by the occupation of the low-energy subspace. Furthermore, in Fig. 6b, we show the corresponding anharmonic part of the free energies. The flat shape of the high smearing (dark red) and the WS$_2$ (gray) curves show that the anharmonicity is strongly reduced compared to the low smearing cases. These observations suggest that the anharmonicities associated with the CDW formation in 1H-TaS$_2$ indeed originate to a large extent from non-linearities in the response of the low-energy electronic system to the external displacement-induced potentials.

Anharmonicities associated with the non-linear low-energy electronic response comprise single-particle and Coulomb contributions. We analyze these contributions diagrammatically in the following for the grand canonical potential $\Omega$:

Model III has the Coulomb contributions accounted for indirectly via the fully screened DFPT deformation-induced potential and the diagrams contributing to anharmonicities in $\Omega$ are of the following types:

$$\Omega^{\mathrm{III}}_{\mathrm{anh}} = \text{[diagram]} + \text{[diagram]} + \dots \tag{17}$$

Model II has explicit Coulomb interaction entering and the diagrammatic content is determined by the approximation used to treat the Coulomb interaction in model II. When solving model II in self-consistent Hartree approximation, we generate terms screening the deformation-induced potential according to Eq. (14). Thus, the anharmonic contributions to the grand potential in model II, $\Omega^{\mathrm{II}}_{\mathrm{anh}}$, contain those diagrams also present in model III but also further ones. For example, at order $\boldsymbol{u}^4$, model II contains a diagram of the form

$$\text{[diagram]} \tag{18}$$

which is not present in model III.

Both the Green's function (not shown here) and total energy or grand canonical potential in models II and III agree at small displacements (by construction) and disagree at higher orders in $u$, and their difference scales with the strength of the Coulomb interaction. Fundamentally, the Green's function of the exact DFT solution contains interaction-mediated anharmonic response to displacements, just like model II does. At the same time, in our current implementation model II only contains Hartree-like diagrams of this kind and lacks other diagram topologies present in the exact DFT solution. These additional diagrams can lead to substantial error cancellation. Thus, it is hard to make general arguments about which model to prefer beyond order $u^2$, given the opaqueness of the underlying DFT exchange-correlation functional. We speculate that cancellations similar to those occurring in second order [73, 76] in $u$ could be also effective in higher orders. In our numerical studies, we find that the total energy curves of model II and III are relatively close for the systems studied here.

## 4 Downfolding-based molecular dynamics

So far we have seen that the downfolded models can reproduce total free energies from DFT. In the following Section 4.1, we assess the computational speed of these models, which ultimately paves the way to enhanced sampling simulations based on MD. As a demonstration of this enhancement, we perform the downfolding-based MD for the example case of monolayer 1H-TaS$_2$ in Section 4.2.

### 4.1 Benchmark of model III against DFT: Force and free energy calculations

To demonstrate the performance gain of model III, we benchmark the calculation of forces and free energies against DFT. For this benchmark, we perform structural relaxations of 1H-TaS$_2$ starting from random displacements $|u_i| < 0.01$ Bohr—to mimic the conditions of a MD simulation step—on different supercells. Durations are averaged over five steps, excluding the first step starting from the initial guess for the density in the DFT case. Calculations are performed on identical machines, using equivalent computational parameters (cf. Appendix C). The results are shown in Fig. 7.

More precisely, we benchmark two implementations of model III: Calculations on finite $k$ meshes, as shown in the previous Section 3, currently require a lot of memory to store the deformation-induced potential in the real-space ($d_{R,R'}$) and reciprocal-space ($d_{q=0,k}$) representations, which limits the system to similar sizes as achievable in DFT (Fig. 7a). Thus, in this section, we instead use a sparse representation, which uses significantly less memory (Fig. 7b), reaching linear scaling with the system size (cf. Ref. [46]), but is currently restricted to $k = 0$, appropriate for large supercells. It also increases the time needed to initialize the program (Fig. 7c, d), which however does not influence the MD simulations. Comparing to the same DFT program we use to obtain the parameters for the downfolded model, i.e., the plane-wave code QUANTUM ESPRESSO [83, 84], we find a speedup of about five orders of magnitude in the downfolding approach for the relevant systems (Fig. 7e, f). Note that our implementation is based on NUMPY and SCIPY [85, 86] and that optimizations both on the *ab initio* and on the model side are possible.

The computational advantage from the non-interacting model III over DFT is easily explained: While DFT relies on the self-consistent solution of the Kohn-Sham system, model III only needs a single matrix diagonalization to solve the Schrödinger equation, thus making it the fastest of all three models. Model I and II on the other hand, incorporate the Coulomb interaction through a self-consistent Hartree algorithm. Assuming a typical number of $\sim 10$ cycles needed for convergence the speedup should be on the order of $10^4$ rather than $10^5$. Most importantly, through downfolding, the matrix of all downfolded models only covers the

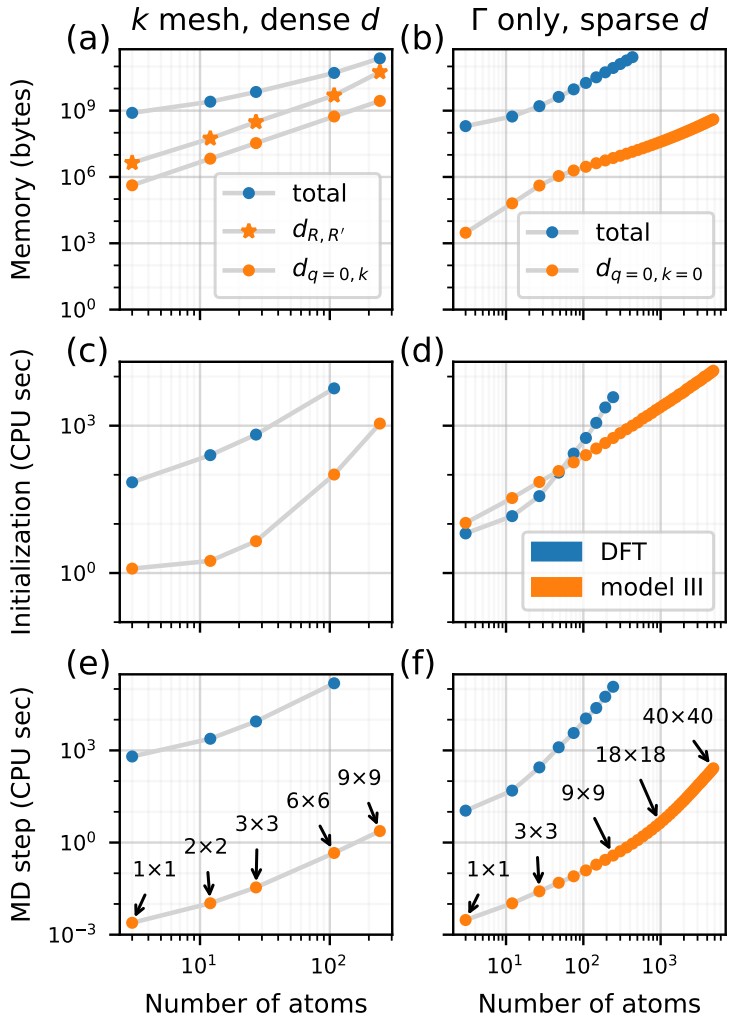

Figure 7: Comparison of (a, b) memory requirements, (c, d) initialization times, and (e, f) durations of energy and forces calculations using QUANTUM ESPRESSO (blue) and our PYTHON implementations of model III (orange) (cf. Appendix C). We consider (a, c, e) $k$ meshes of constant density and (b, d, f) the Γ-only case, for which model III has been implemented using arrays of sparse matrices for the electron-phonon coupling $d_{i\alpha\beta}$. The DFT calculations have been parallelized over plane waves and real-space grids (-nk 1 -nd 1) using 40 CPUs; the model calculations have been run serially. In both cases, Intel Skylake 6148 processors have been used.

low-energy subspace of the electronic structure, as opposed to DFT, whose matrix accounts for low- and high-energy bands.

In fact, most of the time is spent on setting up the Hamiltonian matrix and evaluating the forces [Eq. (11)]. To guarantee that the former is Hermitian and to make the use of sparse matrices more efficient, we have symmetrized $d_{R,R'\alpha\beta} = d^*_{R-R',-R'\beta\alpha}$ and neglected matrix elements smaller than 1 % of the maximum, the effect of which on the free-energy landscape is negligible.

## 4.2 Enhanced sampling simulations based on downfolded model III

We now perform enhanced sampling simulations based on MD with the downfolding scheme defined by model III. To this end, we implemented a PYTHON-based tight-binding solver [74],

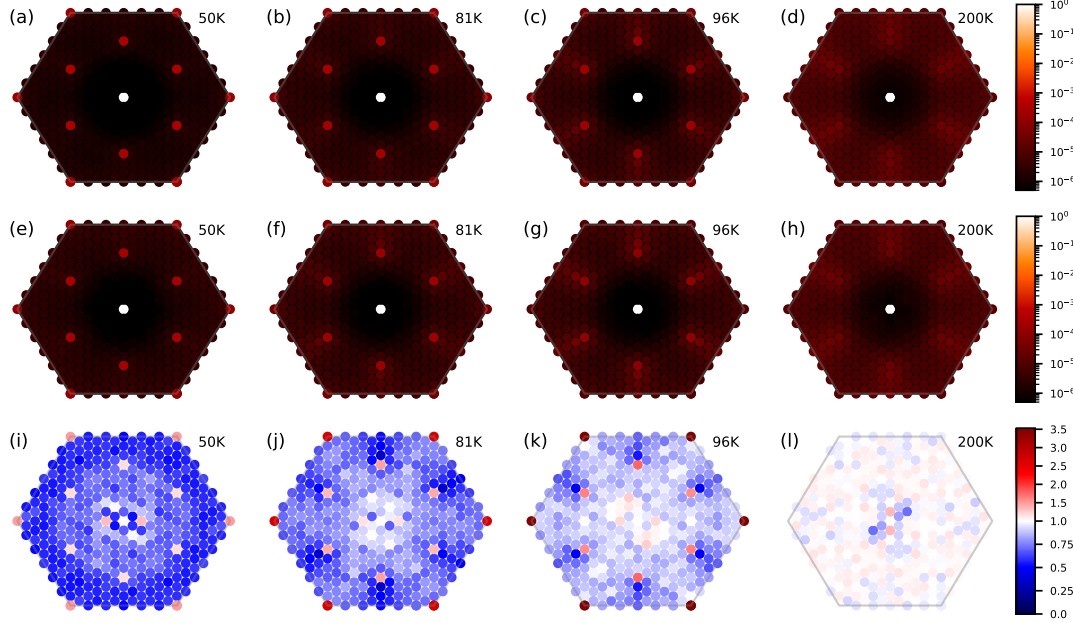

Figure 8: Structure factors $\langle S(\boldsymbol{q})\rangle_T$ [Eq. 19] for 1H-TaS$_2$ on the 18 × 18 supercell. (a–d) Structure factor $S_{\mathrm{CL}}$ from classical MD. (e–h) Structure factor $S_{\mathrm{PI}}$ from PIMD. The peaks at $\boldsymbol{q} = 2/3\,\Gamma\mathrm{M}$ and $\boldsymbol{q} = \mathrm{K}$ for $T = 50$ K are characteristic for the $3 \times 3$ CDW. At higher temperatures, the peaks are broadened and reduced in intensity. (i–l) Ratio of structure factors $S_{\mathrm{CL}}/S_{\mathrm{PI}}$ from classical and path integral MD. A value close to 1 (indicated in white) corresponds to minimal differences between classical and quantum simulations.

which delivers displacement field dependent forces and total free energies to the i-PI (path integral) MD engine [87].

As stated in the previous section, we find a speedup of about five orders of magnitude in the downfolding approach. Thus, the downfolding approaches make larger system sizes and longer time scales well accessible. While for instance Ref. [88] simulates the dynamics of $3 \times 3$ supercells of 1H-NbS$_2$ with *ab initio* MD (AIMD) for time scales of about 6 to 12 ps, the downfolding-based MD allows us to address much larger 18 × 18 supercells for time scales of about 500 ps using a similar amount of CPU hours.[1]

For monolayer 1H-TaS$_2$, we performed classical (and path integral) replica exchange MD simulations (see Appendix D) on the 18 × 18 supercells using 26 replicas (and 10 beads) spanning a temperature range from 50 to 200 K in the canonical (NVT) ensemble. In each MD step $\nu$ we record the position vectors of all nuclei $\boldsymbol{R}_l(\nu, T)$ for all temperatures $T$. Defining the static structure factor

$$S(\boldsymbol{q}) = \frac{1}{N_{\mathrm{at}}^2} \left| \sum_{l=1}^{N_{\mathrm{at}}} e^{-i\boldsymbol{q}\cdot\boldsymbol{R}_l} \right|^2 \tag{19}$$

for a given atomic configuration $\boldsymbol{R}_l$, we obtain the temperature-dependent MD ensemble averaged structure factors $\langle S(\boldsymbol{q})\rangle_T$. We confine the summation to the positions of the tantalum atoms and normalize the structure factor such that $S(\boldsymbol{q} = 0) = 1$. The static structure factor is the frequency integrated version of the dynamic structure factor $S(\boldsymbol{q}) = \hbar \int_{-\infty}^{+\infty} S(\boldsymbol{q}, \omega)d\omega$. Furthermore, it contains both Bragg and all orders of thermal diffuse scattering contributions.

---

[1]The actual simulated times are 430 ps and 930 ps for the path integral (classical) replica exchange MD.

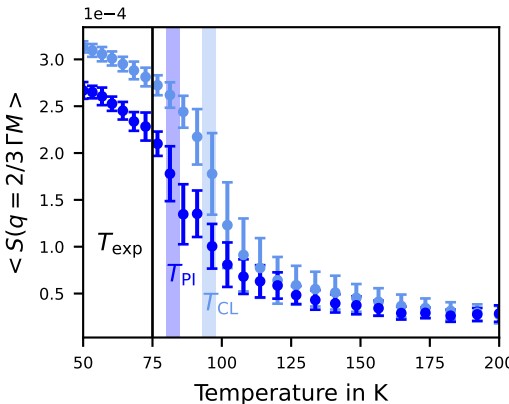

Figure 9: Structure factor $\langle S(\boldsymbol{q} = 2/3\,\Gamma M)\rangle_T$ at the characteristic CDW wavevector $\boldsymbol{q} = 2/3\,\Gamma M$ for the classical MD (light blue) and path integral MD (blue). The effective shift of the PIMD curve toward the experimental value can be attributed to nuclear quantum effects.

The resultant structure factor maps on the first Brillouin zone of 1H-TaS$_2$ are shown in Fig. 8 for temperatures $T = 50, 81, 96, 200$ K.[2] The upper row corresponds to classical MD simulations. At 50 K, we find peaks in the structure factor at $\boldsymbol{q} = 2/3\,\Gamma M$, which are characteristic of the $3 \times 3$ CDW. These peaks broaden and become reduced in intensity upon increasing temperature. Fig. 9 shows the temperature dependence of $\langle S(\boldsymbol{q} = 2/3\,\Gamma M)\rangle_T$ in more detail. We see the aforementioned temperature-induced reduction in $\langle S(\boldsymbol{q} = 2/3\,\Gamma M)\rangle_T$ with an inflection point around $T_{CL} \approx 96$ K. We take this inflection point as the finite system size approximation to the phase transition temperature that would be expected for an infinitely large simulation cell.

While a $3 \times 3$ CDW has been observed in monolayer 1H-TaS$_2$ [89], the exact transition temperature is not known in this system. For the three-dimensional bulk of 2H-TaS$_2$, CDW transition temperatures on the order of $T_{exp} \approx 75$ K have been reported [90–95]. Our classical finite system size estimate exceeds these temperatures by about 25 %. One possible origin of this deviation can be quantum fluctuations in nuclear degrees of freedom.

Therefore, we performed path integral MD (PIMD) replica exchange simulations to assess the influence of nuclear quantum effects on the CDW formation. The PIMD structure factor maps in the middle row of Fig. 8 behave qualitatively similar to the classical counterpart. Their ratio is quantitatively illustrated in the lowest row. The overall area of the Brillouin zone turns from blue to white by heating up the system. Thus, as expected, the classical and quantum simulations agree at high temperatures. However, the CDW fingerprints ($\boldsymbol{q} = 2/3\,\Gamma M$ and $\boldsymbol{q} = K$) clearly increase in intensity and survive at higher temperatures in the classical case. Note that while there is no phonon instability at $\boldsymbol{q} = K$, the corresponding displacements are commensurate with a $3 \times 3$ superstructure and couple anharmonically to the soft modes at $\boldsymbol{q} = 2/3\,\Gamma M$. This explains the high ratios at the Brillouin-zone corners in Fig. 8 (i–k), especially in the vicinity of the transition temperature.

This difference between classical and quantum simulations can be inspected in more detail in Fig. 9. While the qualitative shape of the PIMD curve (dark blue) is similar to the classical MD (light blue) simulation, we find an effective shift of the curve and an inflection point at $T_{PI} \approx 82$ K. Thus, quantum effects can significantly reduce the estimated CDW transition temperature as compared to the classical estimate and lead to a closer match with experiment.

---

[2]We show those $\boldsymbol{q}$ vectors compatible with the periodic boundary conditions on the $18 \times 18$ unit cell.

From these demonstrator calculations it becomes clear that the downfolding-based MD developed in this work opens the gate for precise computational studies of CDW (thermo)dynamics, which were inaccessible in the domain of *ab initio* MD hitherto.

## 5 Conclusions

We presented three downfolding schemes to describe low-energy physics of electron-lattice coupled systems—in particular CDWs—on a similar level of accuracy as full *ab initio* DFT: Model I is based on constrained theories and models II and III are based on unscreening, where model II features explicit Coulomb interactions and model III is effectively non-interacting. The central goal of these downfolding schemes is to reduce the complexity of first-principles electronic structure calculations. This is achieved by mapping the general solid-state Hamiltonian onto minimal quantum lattice models with only a few localized Wannier orbitals per unit cell. The solution of these models is significantly faster than DFT. For model III, we found a speedup of about five orders for the example case of monolayer $1H\text{-}TaS_2$. Despite this enormous speedup and complexity reduction, we demonstrated a quantitative recovery of DFT potential energy surfaces in downfolded models II and III.

As a demonstration, we performed classical and path integral MD simulations using model III of the $1H\text{-}TaS_2$ CDW systems. The downfolding-based speedup opens the gate for enhanced sampling techniques and path integral simulations of nuclear quantum effects on the CDW transition. This makes downfolded models the method of choice for precise computational studies of dynamics and thermodynamics in CDW systems, which were hitherto largely inaccessible to *ab initio* MD.

While we focussed, here, on Born-Oppenheimer MD, the Hamiltonians resulting from downfolded models I–III are generic and likely applicable also when dealing with non-adiabatic phenomena, electron-lattice coupled dynamics in excited electronic states, and situations where strong electron-electron correlations are at play. Due to the explicit account of Coulomb interactions in models I and II, these schemes offer themselves for treatments of situations where electronic interaction effects beyond semilocal DFT are to be included in studies of coupled electron-nuclear dynamics. Furthermore, anharmonic force constants and non-perturbative electron-phonon couplings [96] can be incorporated into the downfolded models to expand the accuracy to even larger lattice distortions.

Future applications of the downfolding schemes developed here might reach to the physics of (nonequilibrium) phase transitions involving CDW order [10–18] or the interplay of correlations and (dis)ordering [19,97] as well as driven quantum systems [4–6]. Beyond potential energy surfaces, the downfolded models can also be used to study the effective electronic structure in the presence of atomic dynamics.

## Acknowledgments

The authors would like to thank Jean-Baptiste Morée for discussions of the RESPACK software package and Bálint Aradi for technical advice.

**Funding information**   We gratefully acknowledge support from the Deutsche Forschungs-gemeinschaft (DFG, German Research Foundation) through RTG 2247 ($QM^3$, Project No. 286518848) (AS, TW), FOR 5249 (QUAST, Project No. 449872909) (TW), EXC 2056 (Cluster of Excellence "CUI: Advanced Imaging of Matter", Project No. 390715994) (TW, SB), EXC 2077 (University Allowance, University of Bremen, Project No. 390741603) (JB), SFB 951

(Project No. 182087777) (MR), and SE 2558/2 (Emmy Noether program) (MAS). AS and TW further acknowledge funding and support from the European Commission via the Graphene Flagship Core Project 3 (grant agreement ID: 881603). JB gratefully acknowledges the support received from the "U Bremen Excellence Chair Program" and from all those involved in the project, especially Lucio Colombi Ciacchi and Nicola Marzari. EvL acknowledges support from the Swedish Research Council (VR) under grant 2022-03090 and from the Crafoord Foundation. We also acknowledge the computing time granted by the Resource Allocation Board and provided on the supercomputer Lise and Emmy at NHR@ZIB and NHR@Göttingen as part of the NHR infrastructure. The calculations for this research were conducted with computing resources under the project hhp00063.

**Data availability** The source code and data associated with this work are available on Zenodo [98].

# A  Free energy calculations of the downfolded models in Hartree approximation

The Coulomb interaction in models I and II renders the electronic Hamiltonian interacting and requires approximate treatments. Here, we solve the interacting Hamiltonian in Hartree approximation, which is the simplest mean-field approximation and as such requires a self-consistency loop.

For the Coulomb interaction, we assume here a density-density type interaction

$$H_{\text{el}}^1 = \frac{1}{2N} \sum_{\boldsymbol{q}\boldsymbol{k}\boldsymbol{k}'\alpha\beta} U_{\alpha\beta}^{\boldsymbol{q}} c_{\boldsymbol{k}+\boldsymbol{q}\alpha}^{\dagger} c_{\boldsymbol{k}'\beta}^{\dagger} c_{\boldsymbol{k}'+\boldsymbol{q}\beta} c_{\boldsymbol{k}\alpha}, \tag{A.1}$$

with $U_{\alpha\beta}^{\boldsymbol{q}}$ being cRPA density-density matrix elements evaluated at momentum transfer $\boldsymbol{q}$.

The Hartree decoupling of Eq. (A.1) reads

$$H_{\text{el}}^1 = \frac{1}{N} \sum_{\boldsymbol{k}\boldsymbol{k}'\alpha\beta} U_{\alpha\beta}^{\boldsymbol{q}=0} \left( c_{\boldsymbol{k}\alpha}^{\dagger} c_{\boldsymbol{k}\alpha} \left\langle c_{\boldsymbol{k}'\beta}^{\dagger} c_{\boldsymbol{k}'\beta} \right\rangle - \frac{1}{2} \left\langle c_{\boldsymbol{k}\alpha}^{\dagger} c_{\boldsymbol{k}\alpha} \right\rangle \left\langle c_{\boldsymbol{k}'\beta}^{\dagger} c_{\boldsymbol{k}'\beta} \right\rangle \right). \tag{A.2}$$

Since the DFT input parameters of models I and II already contain Coulomb contributions, we have to avoid double counting. The hopping terms $t_{\boldsymbol{k}\alpha\beta}^0$ stem from the Kohn-Sham eigenvalues of the undistorted structure, which contain (among others) a Hartree term. Here, we choose $H_{\text{DC}}$ to compensate for the Hartree term of the undistorted structure:

$$H_{\text{DC}} = -\frac{1}{N} \sum_{\boldsymbol{k}\boldsymbol{k}'\alpha\beta} U_{\alpha\beta}^{\boldsymbol{q}=0} \left( c_{\boldsymbol{k}\alpha}^{\dagger} c_{\boldsymbol{k}\alpha} \left\langle c_{\boldsymbol{k}'\beta}^{\dagger} c_{\boldsymbol{k}'\beta} \right\rangle_0 - \frac{1}{2} \left\langle c_{\boldsymbol{k}\alpha}^{\dagger} c_{\boldsymbol{k}\alpha} \right\rangle_0 \left\langle c_{\boldsymbol{k}'\beta}^{\dagger} c_{\boldsymbol{k}'\beta} \right\rangle_0 \right), \tag{A.3}$$

where $\langle \dots \rangle_0$ denotes expectation values obtained for the undistorted structure.

We introduce the Hartree potentials $\overline{U}_\alpha$ and $\overline{U}_\alpha^0$ for the distorted and undistorted structures, respectively,

$$\overline{U}_\alpha = \sum_\beta U_{\alpha\beta}^{\boldsymbol{q}=0} n_\beta \quad \text{and} \quad \overline{U}_\alpha^0 = \sum_\beta U_{\alpha\beta}^{\boldsymbol{q}=0} n_\beta^0, \tag{A.4}$$

where $n_\beta^{(0)} = \frac{1}{N} \sum_{\boldsymbol{k}'\beta} \langle c_{\boldsymbol{k}'\beta}^{\dagger} c_{\boldsymbol{k}'\beta} \rangle_{(0)}$ denotes local orbital occupations. Then, the electronic mean-field Hamiltonian written in the Wannier orbital basis of the supercell reads

$$H_{\text{el}} + H_{\text{el-n}} = \sum_{\boldsymbol{k}\alpha\beta} \left( t_{\boldsymbol{k}\alpha\beta}^0 + \boldsymbol{u} \boldsymbol{d}_{\boldsymbol{k}\alpha\beta} + (\overline{U}_\alpha - \overline{U}_\alpha^0)\delta_{\alpha\beta} \right) c_{\boldsymbol{k}\alpha}^{\dagger} c_{\boldsymbol{k}\beta} - \frac{1}{2} \sum_{\alpha\beta} U_{\alpha\beta}^{\boldsymbol{q}=0} (n_\alpha n_\beta - n_\alpha^0 n_\beta^0). \tag{A.5}$$

This Hamiltonian is solved self-consistently. The converged electronic dispersion $\varepsilon_{kn}$ and occupations $n_\alpha$ are used to determine the free energy:

$$F_{\mathrm{el}} = \frac{2}{N} k_B T \sum_{nk} \ln\left(f\left[-(\varepsilon_{kn} - \mu)/kT\right]\right) + \mu N_{\mathrm{el}} - \frac{1}{2} \sum_{\alpha\beta} U_{\alpha\beta}^{q=0}(n_\alpha n_\beta - n_\alpha^0 n_\beta^0). \tag{A.6}$$

The Coulomb matrix $U_{\alpha\beta}^{q}$ contains one divergent eigenvalue for $q \to 0$, which is associated with the homogeneous charging of the system. Since we are working at fixed system charge, we exclude the divergent contribution of $U_{\alpha\beta}^{q}$. In practice we perform the eigenvector decomposition of Eq. (15) from Ref. [99] and exclude the contribution from the leading eigenvector.

# B  Perturbation expansion of grand potential and free energy

Changes of the grand potential of non-interacting electrons due to atomic displacements can be straightforwardly evaluated using diagrammatic perturbation theory [100]:

$$\Omega = \Omega\big|_0 + \sum_i \Omega_i^{(1)}\big|_0 u_i + \frac{1}{2} \sum_{ij} \Omega_{ij}^{(2)}\big|_0 u_i u_j + \ldots \tag{B.1}$$

$$\equiv \Omega\big|_0 + \text{⟳} + \text{⬡} + \ldots \tag{B.2}$$

Without loss of generality, we consider $q = 0$ in Eq. (9) and drop the corresponding subscript.

In first order, we then have

$$\Omega^{(1)} = \frac{kT}{N} \sum_{kn\nu} d_{knn} \frac{1}{i\omega_\nu - \varepsilon_{kn} + \mu} = \frac{1}{N} \sum_{kn} d_{knn} f(\varepsilon_{kn} - \mu), \tag{B.3}$$

with the Matsubara frequency $\omega_\nu = (2\nu + 1)\pi kT$.

In second order, we have

$$\Omega^{(2)} = \frac{kT}{N} \sum_{kmn\nu} d_{kmn} \frac{1}{i\omega_\nu - \varepsilon_{km} + \mu} \frac{1}{i\omega_\nu - \varepsilon_{kn} + \mu} d_{knm}^T \tag{B.4}$$

$$= \frac{1}{N} \sum_{kmn} d_{kmn} \frac{f(\varepsilon_{km} - \mu) - f(\varepsilon_{kn} - \mu)}{\varepsilon_{km} - \varepsilon_{kn}} d_{knm}^T. \tag{B.5}$$

We deliberately have omitted the superscript zero from $\varepsilon_{kn}$ [cf. Eq. (3)] as in our models with linear electron-phonon coupling these formulas also hold for $u \neq 0$ as long as $d$ is represented in the electronic eigenbasis.

The number of electrons $N_{\mathrm{el}}$ is typically conserved in DFT and MD calculations, so we are instead interested in the canonical ensemble and the free energy

$$F(N_{\mathrm{el}}) = \Omega(\mu(N_{\mathrm{el}})) + \mu(N_{\mathrm{el}})N_{\mathrm{el}}. \tag{B.6}$$

Its first derivative with respect to displacements is

$$F_i^{(1)} = \frac{dF}{du_i} = \frac{\partial \Omega}{\partial u_i} + \left[\frac{\partial \Omega}{\partial \mu} + N_{\mathrm{el}}\right]\frac{d\mu}{du_i} = \Omega_i^{(1)}, \tag{B.7}$$

since $\partial \Omega / \partial \mu = -N_{\mathrm{el}}$. In other words, the expression for the forces [cf. Eq. (11)] is the same in the canonical and the grand-canonical ensemble.

For the unscreening of the force constants (cf. Section 2.1), we also need access to the second derivative of the free energy at constant electron density,

$$F_{ij}^{(2)} = \frac{dF_i^{(1)}}{du_j} = \frac{\partial \Omega_i^{(1)}}{\partial u_j} + \frac{\partial \Omega_i^{(1)}}{\partial \mu}\frac{d\mu}{du_j}.$$

(B.8)

Expectedly, the first term on the right is $\Omega_{ij}^{(2)}$ from Eq. (B.5). Here, the second term does not vanish, at least not for monochromatic perturbations with $\boldsymbol{q} = 0$ [101]. The change of the chemical potential upon atomic displacements follows from the electron conservation,

$$0 \overset{!}{=} \frac{dN_{el}}{d\boldsymbol{u}} = \frac{1}{N}\frac{d}{d\boldsymbol{u}}\sum_{\boldsymbol{k}n} f(\varepsilon_{\boldsymbol{k}n} - \mu) = -\frac{1}{N}\sum_{\boldsymbol{k}n}\Big[\boldsymbol{d}_{\boldsymbol{k}nn} - \frac{d\mu}{d\boldsymbol{u}}\Big]\delta(\varepsilon_{\boldsymbol{k}n} - \mu),$$

(B.9)

with $\delta(\varepsilon) = -df(\varepsilon)/d\varepsilon$. We have used the Hellmann-Feynman theorem,

$$\frac{d\varepsilon_{\boldsymbol{k}n}}{d\boldsymbol{u}} = \frac{d}{d\boldsymbol{u}}\langle \boldsymbol{k}n|H_{el}^0 + H_{el\text{-}n}|\boldsymbol{k}n\rangle = \langle \boldsymbol{k}n|\frac{d}{d\boldsymbol{u}}(H_{el}^0 + H_{el\text{-}n})|\boldsymbol{k}n\rangle \equiv \boldsymbol{d}_{\boldsymbol{k}nn}.$$

(B.10)

Note that here the matrix element of the deformation-induced potential $\boldsymbol{d}_{\boldsymbol{k}nn}$ is represented in the basis of eigenstates $|\boldsymbol{k}n\rangle$ of the *perturbed* Hamiltonian. Rearranging Eq. (B.9) shows that the change of the chemical potential is nothing but the Fermi surface (FS) average of the intraband deformation-induced potential,

$$\frac{d\mu}{d\boldsymbol{u}} = \frac{\sum_{\boldsymbol{k}n}\boldsymbol{d}_{\boldsymbol{k}nn}\delta(\varepsilon_{\boldsymbol{k}n} - \mu)}{\sum_{\boldsymbol{k}n}\delta(\varepsilon_{\boldsymbol{k}n} - \mu)} \equiv \langle \boldsymbol{d}_{\boldsymbol{k}nn}\rangle_{FS}.$$

(B.11)

From Eq. (B.3), we can also readily evaluate

$$\frac{\partial \boldsymbol{\Omega}^{(1)}}{\partial \mu} = \frac{1}{N}\sum_{\boldsymbol{k}n}\boldsymbol{d}_{\boldsymbol{k}nn}\delta(\varepsilon_{\boldsymbol{k}n} - \mu) \equiv \rho(\mu)\langle \boldsymbol{d}_{\boldsymbol{k}nn}\rangle_{FS},$$

(B.12)

where $\rho$ is the electronic density of states per unit cell. Inserting Eqs. (B.11) and (B.12) into Eq. (B.8) yields

$$\Delta C_{ij}^{III} = \Omega_{ij}^{(2)} + \rho(\mu)\langle d_{i\boldsymbol{k}nn}\rangle_{FS}\langle d_{j\boldsymbol{k}nn}\rangle_{FS}.$$

(B.13)

The first term are the force constants in the grand canonical ensemble, i.e., at constant chemical potential. The second term is the correction for going from the grand canonical to the canonical ensemble.

## C Computational parameters for DFT

All DFT and DFPT calculations are carried out using QUANTUM ESPRESSO [83, 84]. The modification that is required for cDFPT is described in detail in Ref. [64]. For the transformation of the electronic energies and electron-phonon couplings to the Wannier basis, we use WANNIER90 [102] and the EPW code [103–105]. The cRPA Coulomb interaction was calculated using RESPACK [106]. In the following, we will list the specific DFT and DFPT parameters for each material individually:

**1H-TaS$_2$** Hartwigsen-Goedecker-Hutter (HGH) pseudopotentials [107, 108]; $18 \times 18 \times 1$ $\boldsymbol{k}$ mesh and $6 \times 6 \times 1$ $\boldsymbol{q}$ mesh for unit cell; Fermi-Dirac smearing of 5 mRy (Gaussian smearing of 0.1 Ry for Fig. 7b, d, f); energy convergence threshold of $10^{-15}$ Ry ($10^{-8}$ Ry per unit cell for Fig. 7); lattice constant of 3.39 Å. The cRPA Coulomb interaction has been calculated on a $32 \times 32 \times 1$ $\boldsymbol{q}$ mesh taking 80 electronic bands into account.

**1H-NbS$_2$** HGH pseudopotentials [107,108]; $18 \times 18 \times 1$ $\boldsymbol{k}$ mesh and $6 \times 6 \times 1$ $\boldsymbol{q}$ mesh for unit cell; Fermi-Dirac smearing of 3 mRy; lattice constant of 3.34 Å.

**1H-WS$_2$** HGH pseudopotentials [107,108]; $18 \times 18 \times 1$ $\boldsymbol{k}$ mesh and $6 \times 6 \times 1$ $\boldsymbol{q}$ mesh for unit cell; Fermi-Dirac smearing of 5 mRy; lattice constant of 3.23 Å.

**1T-TiSe$_2$** Ultrasoft pseudopotential [109] from the SSSP library [110,111]; $18 \times 18 \times 1$ $\boldsymbol{k}$ mesh and $6 \times 6 \times 1$ $\boldsymbol{q}$ mesh for unit cell; Fermi-Dirac smearing of 5 mRy; lattice constant of 3.54 Å.

**Carbon chain** Optimized norm-conserving Vanderbilt pseudopotential (ONCVPSP) [112] from the PSEUDODOJO library [113]; $200 \times 1 \times 1$ $\boldsymbol{k}$ mesh and $20 \times 1 \times 1$ $\boldsymbol{q}$ mesh for unit cell; Fermi-Dirac smearing of 5 mRy; lattice constant of 1.30 Å.

In all cases, we have applied the Perdew-Burke-Ernzerhof (PBE) functional [114], set the plane-wave cutoff to 100 Ry, and minimized forces and pressure in the periodic directions to below 1 μRy/Bohr and 0.1 kbar. We have used a unit-cell dimension of 15 Å to separate images in the non-periodic directions.

## D  Replica exchange

In order to characterize the CDW phase-transition, we employed replica exchange molecular dynamics (REMD) and replica exchange path integral molecular dynamics [87] (PI-REMD), as implemented in the i-PI code. For the $18 \times 18$ 1H-TaS$_2$ supercell, we ran NVT simulations of 26 replicas in parallel that differed in the ensemble temperature. We covered a temperature range between 50 and 200 K. In the PI-REMD simulations, each temperature replica was represented by ten imaginary-time replicas (commonly called "beads" in the ring-polymer representation). This amount of beads proved to be converged within 1 meV/atom for the potential and quantum kinetic energy at the lowest temperature of 50 K. We note that due to the high dimensionality of the system, enhanced by the use of many imaginary-time replicas, the PI-REMD simulations with 26 temperature replicas in this range was not efficient in terms of the frequency of replica swaps, while the REMD simulations were.

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
