# Peer review of "Ab initio electron-lattice downfolding: potential energy landscapes, anharmonicity, and molecular dynamics in charge density wave materials"

_SciPost Physics, doi:SciPost Phys. 16, 046 (2024)_

## Round 2 · Referee Report · Anonymous (Referee 1) · 2023-10-15

Strengths

  1. the approaches here presented and valuable and interesting
  2. the results are useful and well supported
  3. the paper is clear and convincing

Weaknesses

  1. no weak point

Report

In this paper the Authors present three different downfolding schemes for approching phonon anharmonicity and phase transitions in compounds with charge-density-waves (CDWs).
They test the validity of these scheme against few selected representative materials, showing their efficiency.
The paper is sound, well written and the results are interesting and useful.
I gladly recommend publication.

Requested changes

  1. no requested change

---

## Round 2 · Referee Report · Anonymous (Referee 2) · 2023-10-21

Strengths

  1. Well-written
  2. Excellent implementation
  3. Accurate and efficient methodology

Weaknesses

More applications regarding the temperature-dependent effect of the CDW on the electronic structure? (and some other minor points in the report)

Report

n this work, Schobert et al performed an interesting study on CDW systems for exploring anharmonic potential energy surfaces accurately and efficiently using downfolded models to DFT and ab initio MD. They studied three different downfolding strategies and considered four different systems (1H-TaS 2 , 1T-TiSe 2 , 1H-NbS 2 , and a carbon chain) to find that the models are very beneficial reducing by a large factor the complexity of ab initio MD electronic structure calculations. The work is very interesting and deserves publication as it indeed offers a route for speeding up path integral simulations of quantum nuclear effects which are rather cumbersome to be applied for extended systems. Below I provide some other points for consideration:

  1. Typo p.12: "for the case example of monolayer".

  2. p.4 : "since DFT calculations with large supercells are prohibitively expensive" Maybe the authors could quantify further the supercell size and explain why is needed in more detail.

  3. Can the authors comment in more detail on the relative comparison of the models with respect to computational efficiency?

  4. Eq. (19): (a) Are the atomic scattering amplitudes missing from the equation of the structure factor? How is this relation related to diffuse scattering? (b) How many scattering wavevectors are used for the sampling? To me it looks to be a coarse grid. (c) How many configurations are needed to obtain a converged average ? (d) What is the nature of CDW peaks? Is it quasi-elastic or inelastic? (e) Can the authors comment whether this term includes contributions to diffuse scattering both from coupled and independent lattice vibrations (one-phonon and multiphonon scattering)?

  5. Is there a double definition of N? Does it represent the number of k-points and number of atoms? I think it's better for the authors to make sure that no definitions with the same symbol appear in the text.

  6. Let me also suggest a new paper for the EPW code: https://doi.org/10.1038/s41524-023-01107-3

  7. Can the authors comment if anharmonic temperature-dependent lattice constants C^{DFT} can be used in Models II and III? Will it be more appropriate? More, will it be more beneficial and more consistent if the deformation potential is computed with nonperturbative supercell calculations to electron-phonon coupling (see for example: https://doi.org/10.1088/1367-2630/aaf53f)? I am simply sharing thoughts here.

  8. Will the first-principles calculations (inputs and outputs) uploaded to a database or the Python codes on open-source platforms?

  9. I think the work has some more space to be expanded for example in the calculation of other thermal averages related to the electronic structure.

  • validity: high
  • significance: high
  • originality: high
  • clarity: top
  • formatting: excellent
  • grammar: excellent

Author:  Arne Schobert  on 2024-01-17  [id 4258]

(in reply to Report 2 on 2023-10-21)

The new manuscript has been resubmitted. Please find the "reply.pdf" document below that answers directly to these questions and includes a formal list of changes.

Attachment:

reply.pdf

---

## Round 3 · Referee Report · Anonymous (Referee 2) · 2024-1-22

Report

The authors have addressed my comments and the manuscript is ready for publication.

---

## Round 3 · Author Response

Dear editor,
We would like to thank the referees for their reports, and we are pleased to see that they are overall positive
about our manuscript. The questions in the report mainly deal with the computational efficiency of the method,
its possible extensions, and the details of the structure factor. We have improved our manuscript to clarify these
points. Below, we discuss the points made in the report and the requested changes one by one. With these
changes, listed at the end of the document, we believe our manuscript is suitable for publication in SciPost
Physics.
Yours sincerely,
The authors

---

## Round 3 · List of Changes

The point-by-point list of changes is included in the "reply.pdf" document that was uploaded as an answer to the referees.

---

## Editorial Decision

published